# AI-Driven Segmentation and Automated Analysis of the Whole Sagittal Spine from X-ray Images for Spinopelvic Parameter Evaluation

**DOI:** 10.3390/bioengineering10101229

**Published:** 2023-10-20

**Authors:** Sang-Youn Song, Min-Seok Seo, Chang-Won Kim, Yun-Heung Kim, Byeong-Cheol Yoo, Hyun-Ju Choi, Sung-Hyo Seo, Sung-Wook Kang, Myung-Geun Song, Dae-Cheol Nam, Dong-Hee Kim

**Affiliations:** 1Department of Orthopaedic Surgery, Institute of Medical Science, Gyeongsang National University Hospital and Gyeongsang National University School of Medicine, Jinju 52727, Republic of Korea; songsangyoun@naver.com (S.-Y.S.); majestyno1@naver.com (M.-S.S.); kcw_blue@naver.com (C.-W.K.); 2Deepnoid. Inc., Seoul 08376, Republic of Korea; kim01414@deepnoid.com (Y.-H.K.); bcyoo79@deepnoid.com (B.-C.Y.); hjchoi@deepnoid.com (H.-J.C.); 3Department of Biomedical Research Institute, Gyeongsang National University Hospital, Jinju 52727, Republic of Korea; sunghyo.seo@gmail.com; 4Precision Mechanical Process and Control R&D Group, Korea Institute of Industrial Technology, Seoul 06211, Republic of Korea; swkang@kitech.re.kr; 5Department of Orthopaedic Surgery, College of Medicine, Inha University Hospital, Incheon 22212, Republic of Korea; piano10000@naver.com

**Keywords:** artificial intelligence, deep learning, automated analysis, whole-spine lateral radiographs, spinopelvic parameters

## Abstract

Spinal–pelvic parameters are utilized in orthopedics for assessing patients’ curvature and body alignment in diagnosing, treating, and planning surgeries for spinal and pelvic disorders. Segmenting and autodetecting the whole spine from lateral radiographs is challenging. Recent efforts have employed deep learning techniques to automate the segmentation and analysis of whole-spine lateral radiographs. This study aims to develop an artificial intelligence (AI)-based deep learning approach for the automated segmentation, alignment, and measurement of spinal–pelvic parameters through whole-spine lateral radiographs. We conducted the study on 932 annotated images from various spinal pathologies. Using a deep learning (DL) model, anatomical landmarks of the cervical, thoracic, lumbar vertebrae, sacrum, and femoral head were automatically distinguished. The algorithm was designed to measure 13 radiographic alignment and spinal–pelvic parameters from the whole-spine lateral radiographs. Training data comprised 748 digital radiographic (DR) X-ray images, while 90 X-ray images were used for validation. Another set of 90 X-ray images served as the test set. Inter-rater reliability between orthopedic spine specialists, orthopedic residents, and the DL model was evaluated using the intraclass correlation coefficient (ICC). The segmentation accuracy for anatomical landmarks was within an acceptable range (median error: 1.7–4.1 mm). The inter-rater reliability between the proposed DL model and individual experts was fair to good for measurements of spinal curvature characteristics (all ICC values > 0.62). The developed DL model in this study demonstrated good levels of inter-rater reliability for predicting anatomical landmark positions and measuring radiographic alignment and spinal–pelvic parameters. Automated segmentation and analysis of whole-spine lateral radiographs using deep learning offers a promising tool to enhance accuracy and efficiency in orthopedic diagnostics and treatments.

## 1. Introduction

The prevalence of degenerative spinal diseases in Korea is increasing annually by 3–4% due to aging and sedentary lifestyles [1]. Common degenerative spinal diseases such as disc herniation and spinal stenosis have been on the rise, with an increasing number of patients seeking treatments. According to data from the Korean National Health Insurance Service (KNHIS), there was a significant increase of approximately 1.12 million patients diagnosed with spinal disorders in South Korea between 2015 and 2019. It is important to highlight that the most rapidly increasing demographic comprises individuals in their 20s. Although numerous studies have investigated treatment and prevention strategies for spinal disorders, recent research has shifted its focus towards the role of spinal alignment in these conditions, emphasizing the significance of malalignment factors, such as spinal and pelvic parameters [2,3].

The significance of spinal alignment has gained increasing recognition as a quantitative assessment of spinal health, given its association with various spinal disorders, including idiopathic scoliosis, adult spinal deformity, and degenerative sagittal imbalance. Malalignment, along with compensatory mechanisms to maintain an upright posture, can impose additional stress on critical spinal, pelvic, and hip joints, potentially leading to joint pain and arthritis [4,5].

Currently, in clinical practice, various landmarks are used to measure key angles and distances, and radiographic evaluation of spinal–pelvic alignment is performed prior to surgery by comparing the measured values with the predetermined alignment goals [6,7]. However, such manual measurements can vary depending on the examiner and the institution’s radiographic imaging protocols, necessitating a correction process. In this context, a clinically developed system for accurate spinopelvic parameter assessment is currently lacking.

With the rapid advancement of artificial intelligence (AI) and its increasing integration into various fields, there is potential to develop AI-driven solutions for more precise and reliable spinopelvic parameter evaluations [8,9]. By leveraging the capabilities of AI, such as machine learning and computer vision, it may be possible to enhance the consistency and accuracy of measurements, reduce variability between examiners and institutions, and ultimately improve patient outcomes. Furthermore, AI-based systems could offer real-time analysis and support decision-making processes, enabling clinicians to more effectively diagnose, treat, and manage spinal disorders [10].

Despite the potential benefits of AI in the context of spinopelvic parameter assessment, several challenges persist. Currently, there is a scarcity of AI models that have been trained on large-scale radiographic datasets, which limits their capacity to accurately evaluate spinal alignment. Additionally, the evidence supporting the inter-rater reliability of AI-driven systems remains insufficient, further hindering their widespread adoption in clinical practice.

Based on previous studies and the literature, we hypothesize that the proposed algorithm for automated whole-spine–pelvic detection and segmentation, which utilizes deep learning techniques and is trained on extensive radiographic datasets, will demonstrate high accuracy and efficiency in generating radiological alignment and spinopelvic parameters from whole-spine lateral radiographs. The accuracy of the algorithm will be assessed by evaluating the inter-rater reliability between the deep learning (DL) model and senior spine specialists of orthopedic surgeons and orthopedic residency, using the within-class correlation coefficient (ICC). We further hypothesize that the proposed algorithm will offer a promising tool for improving the accuracy and efficiency of diagnosis and treatment in orthopedic surgery, with potential applications in various spinal disease etiologies. If successful, the proposed AI algorithm could revolutionized the field of orthopedic surgery and spinal disorder diagnosis. It will make assessments more accurate, consistent, and efficient. This advancement can lead to better patient outcomes, fewer surgical complications, and a more streamlined process for medical professionals in evaluating and treating spinal disease.

## 2. Materials and Methods

### 2.1. Study Design

From January 2015 to April 2022, a total of 932 consecutive whole-spine lateral plain radiographs were retrospectively reviewed and annotated under the approval of the institutional review board (IRB No. 202205007002).

### 2.2. Datasets

This study employed instance segmentation to automatically detect and segment whole-spine–pelvic structures in whole-spine lateral radiographs. The dataset consisted of a total of 932 subjects, with 748 subjects used for training, 94 for validation, and 90 for testing. In this study, cases were excluded from the analysis if they involved patients with previous surgeries, pregnant individuals, inadequate radiographs, minors with underdeveloped skeletons, or instances of degenerative disk disease where the corners of a vertebra were challenging to observe.

### 2.3. AI Model Architecture

In this study, the data were pre-processed by converting them into a COCO-style dataset without contrast enhancement, zero padding, or resizing to 2048 × 2048 pixels. Additionally, min-max normalization was applied. Three distinct models with ResNet50-FPN backbones and varying numbers of hidden layers were trained and evaluated utilizing the COCO API. Evaluation metrics included Intersection over Union (IoU) and mean average precision (mAP) scores. Score thresholds were established for both detection and segmentation to exclude predictions below a specific probability.

The Mask R-CNN model is comprised of three networks: a backbone network for feature extraction from the input image, a region proposal network (RPN), and a mask prediction network [11]. ResNet with 50 layers and a Feature Pyramid Network (FPN) was employed as the backbone network for this model. FPN generates multiple feature maps with different scales, where the lowest level has the highest spatial resolution corresponding to the input image [12,13]. Higher levels of feature maps have lower spatial resolution but contain more semantic features. To construct the highest semantic feature maps, each feature map is unsampled to the scale of the lower-level map and subsequently merged.

The RPN predicts regions in the input image that are likely to contain objects. It utilizes anchors with various sizes and aspect ratios to output rectangular region proposals as sets of coordinates and corresponding objectless scores. The mask prediction network consists of two branches: a region proposal branch and a mask proposal branch. The region proposal branch includes fully connected layers for classification and regression, outputting the object class and region proposal. For the mask proposal branch, a fully convolutional network (FCN) is typically implemented to process and predict binary object instance masks within each region. Pixels with a value of 0 indicate background, while those with a value of 1 indicate the object instance [14].
$$L\left(p,u,t^{u},v\right)=L_{cls}\left(p,u\right)+\lambda^{\prime }L_{reg}\left(t^{u},\;v\right)$$
$$L\left(\{p_{i}),\;\left\{t_{i}\right\}\right)=\frac{1}{N_{cls}}\underset{i}{\sum }L_{cls}\left(p_{i},\;p_{i}^{*}\right)+\lambda \frac{1}{N_{reg}}\underset{i}{\sum }p_{i}^{*}L_{reg}\left(t_{i},t_{i}^{*}\right)$$

Two loss functions, R-CNN loss and RPN loss, are employed for training the Mask R-CNN model [15]. The model’s hyperparameters included an Adam optimizer, a batch size of 4, and 50 epochs, which demonstrated optimal performance on the test data [16] (Figure 1).

### 2.4. Spinopelvic Parameter Measurement

Thirteen radiological alignment and spinopelvic parameters, including Cervical 2 Incidence (C2I), Cervical 2 Slope (C2S), Cervical 2 to 7 Lordosis (C27L), Thoracic 1 Incidence (T1I), Thoracic 1 Slope (T1S), Thoracic Kyphosis (TK), Lumbar 1 Incidence (L1I), Lumbar Lordosis (LL), Sacral Slope (SS), Pelvic Tilt (PT), Pelvic Incidence (PI), Cervical 2 to 7 Sagittal Vertical Axis (C2-7 SVA), and Cervical 7 Sagittal Vertical Axis (C7 SVA), were automatically generated from the whole-spine lateral radiograph using the deep learning model that distinguished anatomical landmarks of the cervical, thoracic, lumbar, sacral, and femoral heads (Figure 2).
Cervical 2 Incidence (C2I) is the angle between a line connecting the center of the femoral heads (yellow dashed line) to the midpoint of the sacral superior endplate and the line perpendicular to the C2 inferior endplate (brown dashed line).Cervical 2 Slope (C2S) denotes the angle between the horizontal line (red dashed line) and the inferior endplates of C2.Cervical 2 to 7 Lordosis (C27L) refers to the angle between the inferior endplate of C2 and the inferior endplate of C7.Thoracic 1 Incidence (T1I) denotes the angle between a line connecting the center of the femoral heads (yellow dashed line) to the midpoint of the sacral superior endplate and the line perpendicular to the T1 superior endplate (brown dashed line).Thoracic 1 Slope (T1S) represents the angle between the horizontal line (red dashed line) and the superior endplates of T1.Thoracic Kyphosis (TK) refers to the angle between the superior endplate of T1 and the superior endplate of L1.Lumbar 1 Incidence (L1I) is the angle between a line connecting the center of the femoral heads (yellow dashed line) to the midpoint of the sacral superior endplate and the line perpendicular to the L1 superior endplate (brown dashed line).Lumbar Lordosis (LL) represents the angle between the superior endplate of L1 and the superior endplate of S1.Sacral Slope (SS) is the angle between the sacral superior endplate and the horizontal line (red dashed line).Pelvic Tilt (PT) refers to the angle between a line connecting the center of the femoral heads (yellow dashed line) to the midpoint of the sacral superior endplate and the vertical line (blue dashed line).Pelvic Incidence (PI) is the angle formed by a line connecting the center of the femoral heads (yellow dashed line) to the midpoint of the sacral superior endplate and the line perpendicular to the sacral superior endplate (brown dashed line). The green lines depict the parallel lines corresponding to each endplate.Cervical 2 to 7 Sagittal Vertical Axis (C2-7 SVA) is a parameter that measures the horizontal distance between a plumb line dropped from the center of the C2 vertebral body (blue dashed line) and the posterosuperior corner of the C7 vertebral body, within the spinal segment that includes the second to seventh cervical vertebrae.Cervical 7 Sagittal Vertical Axis (C7 SVA) measures the horizontal distance between the posterosuperior corner of S1 and the plumb line dropped from the center of the C7 vertebral body (blue dashed line).

### 2.5. Graphical User Interface (GUI)

In this study, we employed a software application developed using Python 3.9.7 and the Tkinter library for the graphical user interface. Upon opening an image within the software, it is processed by our pre-trained model, which identifies and predicts the masks for the vertebrae, sacrum, and femur structures. To measure the relevant parameters, we approximated the shape of each generated mask and subsequently determined the two upper corner points using the OpenCV library, a widely recognized computer vision toolkit (Figure 3).

### 2.6. Intraclass Correlation Coefficient (ICC) and Statistical Analyses

To evaluate the agreement between measurers (specialists; residents—3rd year trainee; and AI), the Intraclass Correlation Coefficient (ICC) was used as an indicator. ICC values range from 0 to 1, with 0 indicating no agreement and 1 indicating perfect agreement. In this study, ICC(2,1), which utilizes a 2-way random effect, absolute agreement, and single rater/measurement, was employed. There are no absolute standards for ICC, but it is generally classified as poor (<0.40), fair (0.40–0.60), good (0.60–0.75), and excellent (0.75–1.00). The agreement between the values measured by residents and AI models was calculated using the specialist as the reference standard. The ‘pingouin’ package in Python was employed for calculating ICC. Additionally, the Bland–Altman plot was employed to assess the agreement between the deep learning model, specialists and residents. The agreement was further evaluated by comparing the measurements from residents and AI models with those from specialists. To determine the statistical significance of differences between groups, either the *t*-test or analysis of variance (ANOVA) was utilized. This approach allowed for a comprehensive understanding of the agreement between the various measurers and facilitated the identification of any potential discrepancies or areas of improvement.

## 3. Results

### 3.1. AI Detection and Segmentation

The best model was selected based on the highest AP0.75 score for segmentation, and all models were trained with a batch size of 4 and a learning rate of 0.0005 with 0.91 decay for 30 epochs. The results showed that the models achieved high accuracy in detecting and segmenting spinal structures in whole-spine lateral radiographs.

The performance of the detection model was assessed using three variations of the ResNet50 backbone model: HL256, HL512, and HL1024. The COCO API was used to calculate the AP scores for validation and test data sets. The HL256 model achieved AP scores of 0.626 and 0.604 for validation and test data sets, respectively, with an AP0.75 score of 0.747 and 0.693 for validation and test data sets, respectively. The HL512 model demonstrated slightly better performance than the HL256 model, with AP scores of 0.629 and 0.618 for validation and test data sets, respectively, and an AP0.75 score of 0.74 and 0.73 for validation and test data sets, respectively. The HL1024 model exhibited similar performance to the HL256 model, with AP scores of 0.628 and 0.612 for validation and test data sets, respectively, and an AP0.75 score of 0.727 and 0.714 for validation and test data sets, respectively. The AP scores for small, medium, and large objects were also reported. Model weights were selected based on the best AP0.75 score of segmentation.

The evaluation scores of the segmentation model were obtained for three different variations of the ResNet50 backbone model: HL256, HL512, and HL1024. The AP scores, as measured by COCO API, were calculated for validation and test data sets. For the HL256 model, the AP scores were 0.614 and 0.586 for validation and test data sets, respectively, with an AP0.75 score of 0.724 and 0.681 for validation and test data sets, respectively. The HL512 model showed slightly better performance than the HL256 model, with AP scores of 0.624 and 0.602 for validation and test data sets, respectively, and an AP0.75 score of 0.718 and 0.706 for validation and test data sets, respectively. The HL1024 model showed similar performance to the HL256 model, with AP scores of 0.610 and 0.590 for validation and test data sets, respectively, and an AP0.75 score of 0.721 and 0.670 for validation and test data sets, respectively. The AP scores for small, medium, and large objects were also reported. The model weights were selected based on the best AP0.75 score of segmentation (Figure 4).

### 3.2. Angle Measurement

The success rate of angle measurement based on the segmentation result was evaluated using the validation and test datasets. The hidden layers used were based on the weights of the best AP0.75 score of segmentation. An IoU threshold of 0.80 was applied, and three validation cases and seven test cases failed on all models.

### 3.3. Graphical User Interface (GUI)

We developed a program using the Tkinter library, which is the default GUI library for Python. The program allows users to open an image, which is then passed to our trained model for analysis. The model predicts the masks for the vertebrae, sacrum, and femur. To extract the spinopelvic parameters, we approximated the shape of the mask and used the OpenCV library to obtain the two upper corners. To compare the results with the Korean standard, we used a color-coded system in our program. If the results were within the normal range, they were displayed in blue, and if they were outside the normal range, they were displayed in red. This allowed users to easily identify any abnormalities in the measurements. We also ensured that the program displayed the specific measurement results for each parameter, which enabled users to confirm how the measurements were taken. Additionally, we incorporated a feature to allow users to save the results for future reference (Figure 5).

The process of extracting all the spinopelvic parameters using our program took an average of 3.8 s, with a standard deviation of 0.8 s.

### 3.4. Intraclass Correlation Coefficient (ICC)

The intra-class correlation coefficient (ICC) was utilized to quantify the level of agreement between measurements, with a value of 1 representing perfect agreement, while 0 indicates the lack of agreement. The 95% confidence interval (CI) was calculated to establish a range of values, within which we can be 95% certain that the true ICC value lies.

Utilizing the key points identified by the proposed technique, 13 sagittal parameters were computed. To assess the consistency between the performance of the developed method and the manual measurements conducted by specialists, we analyzed the correlation coefficient and the coefficient of determination between the two sets of measurements, as displayed in Table 1.

Based on the results presented in the table, the comparison between Specialist vs. Resident and Specialist vs. AI demonstrates the potential effectiveness of utilizing AI for specific measurements. In terms of R-squared values, the Specialist vs. AI comparison reached the highest value of 0.956 for the C2-7 measurement and the lowest of 0.066 for Time (s). For Specialist vs. Resident, the highest R-squared value was 0.926 for the C2-7 measurement, and the lowest was 0.085 for the PT measurement. The AI method seemed to face difficulties in measuring T1-related parameters, with a correlation coefficient of 0.656 for T1S and 0.794 for T1I. This difficulty was also reflected in the mean absolute error (MAE) and standard deviation of absolute error (STD of AE), with the lowest MAE value being 2.112° for the C2-7 measurement and the largest error stemming from Time (s) at 736.245. Taking the correlation coefficient as an example, the PT parameter shows a value of 0.844 for Specialist vs. AI and a lower value of 0.717 for Specialist vs. Resident. This indicates that the AI’s performance is closer to the Specialist in this aspect. Similarly, the R-squared value for the SS parameter is higher for Specialist vs. AI at 0.631 compared to 0.325 for Specialist vs. Resident. This demonstrates that the AI model has a better agreement with the Specialist’s measurements. In terms of MAE, the PI parameter has a value of 5.689 for Specialist vs. AI, while it is significantly higher at 9.429 for Specialist vs. Resident. This reveals that the AI has a lower average error compared to the Resident when compared to the Specialist. Lastly, the STD of AE value for the SI parameter is lower for Specialist vs. AI at 4.299, as opposed to 7.682 for Specialist vs. Resident. This suggests that AI’s measurements have less variability and are closer to the Specialist’s measurements.

Our findings revealed that the AI quickly measured the angles, while the human was slower. The mean value of the measurements obtained by specialists was 904.15, with a mean error of 204.97 and a standard error of 28.27. On the other hand, the mean value of the measurements obtained by experts was 732.4, with a mean error of 105.1 and a standard error of 16.1. In comparison, the AI system achieved a mean value of 3.73, with a mean error of 0.73 and a standard error of 0.1. It is noteworthy that the AI system exhibited significantly lower mean error and standard error compared to both specialists (Table 2).

The findings indicate that the ICC values for most variables are high, signifying substantial agreement among the different methods. Specifically, C2-7 SVA, C2S, C27L, and L1I have ICC values above 0.85, indicating excellent agreement between the measurements obtained via the different methods. However, some variables show relatively low ICC values, such as SS, which has an ICC value of only 0.27, indicating poor agreement among the different methods. When comparing the measurements obtained by specialists with the reference values, the results indicate that the AI-based method demonstrated good agreement with the specialist measurements for most variables. Specifically, for C2-7 SVA, C2S, C27L, and L1I, the ICC values were above 0.90, indicating excellent agreement between the AI-based measurements and those obtained by specialists. However, for some variables, such as SS and TK, the AI-based measurements exhibited lower ICC values when compared to specialist measurements. For SS, the ICC value was 0.62, indicating moderate agreement, while for TK, the ICC value was 0.74, indicating fair agreement (Table 3).

The Bland–Altman plot was used to visualize the results of the method for all joint parameters, showing the mean difference (red solid line) and the horizontal lines representing the mean difference ± 1.96 standard deviation (SD) (blue dashed line).

For example, for the C2I parameter, the mean difference between specialists and residents was 0.63, and the mean difference between specialists and AI was relatively lower at 0.33. On the other hand, for the TK parameter, the mean difference between specialists and residents was 2.57, and the mean difference between specialists and AI was relatively higher at 4.21.

For C7 SVA, the mean difference between specialists and residents was 5.6, and the mean difference between specialists and AI was relatively lower at 2.58.

For the Time parameter, the mean difference between specialists and residents was 173.71, and the mean difference between specialists and AI was very high at 736.24 (Table 4).

## 4. Discussion

Our study aimed to develop an efficient and automated method for extracting spinopelvic parameters from whole-spine long sagittal X-ray images using artificial intelligence (AI). To achieve this, we designed and implemented a deep learning model capable of accurately detecting and segmenting spinal structures, such as vertebrae, sacrum, and femur. Moreover, we created a user-friendly GUI using the Tkinter library in Python, enabling healthcare professionals to easily input X-ray images and obtain the spinopelvic measurements in a matter of seconds. Overall, our study provides a promising approach to streamlining and improving the accuracy of spinopelvic parameter extraction from X-ray images using AI and a user-friendly interface.

The AI-based method can rapidly analyze spinal alignment changes over time and between individuals, providing a more comprehensive understanding of spinal sagittal alignment. This accumulated knowledge can be utilized to determine the most effective approach for spinal deformity correction, enabling personalized treatment plans for each patient. Deep mining algorithms can compare patients’ spinal alignment data to identify patterns associated with positive and negative prognoses, leading to improved patient outcomes. In this way, AI technology has the potential to revolutionize the field of spinal surgery and provide more effective and personalized treatment options. While the role of AI in spinal diagnostics is undeniably transformative, it is essential to view it as a part the broader diagnostic tool, one that complements, not replaces, clinical expertise. As the field of spinal care evolves, the synergy between AI-derived insights and hands-on clinical experience will make better patient prognosis.

Recent advancements in deep learning technology have led to the development of segmentation techniques for X-ray, CT, and MRI images. However, in the medical field, accuracy in measurement is crucial. Based on the results presented in the table, the proposed AI-based method demonstrated good agreement with specialist measurements for most variables. The ICC values were above 0.90 for C2-7 SVA, C2S, C27L, and L1I, indicating excellent agreement between the AI-based measurements and those obtained by specialists. Our study found an mean error of 3.625 for C2-7L measurements, which is comparable to the mean error of 3.3 for cervical lordosis obtained from an AI model trained on 4546 cervical X-rays [17]. The relatively high error may be attributed to including patients with deformities, those with poorly visible C7 endplates due to obesity, and patients with short necks who may have a different shoulder girdle position in the training process. Despite the inclusion of these prevalent patient groups in clinical practice, it was deemed important to ensure generalizability of the training model. However, future efforts to refine the model to reduce the error range will be necessary.

To achieve high accuracy, extensive image training is essential. In contrast to previous studies, our research utilized over 900 images for training, resulting in an average error of 8.789° for Lumbar Lordosis Angle (LLA) measurements. Another study that trained on 629 individuals for LLA measurements showed a similar level of mean error at 8.055° [17]. Our model also demonstrated lower error compared to a model trained on 493 individuals using EOS, which reported a mean absolute error (MAE) of 11.5° [18]. These findings highlight the importance of utilizing a large dataset for training to achieve higher accuracy in medical imaging applications.

The detection of endplates in the cervicothoracic junctional area is challenging, which has resulted in significant differences in measurements. For instance, the mean difference between specialists and residents was 2.57 for the TK parameter, while the mean difference between specialists and AI was relatively higher at 4.21. Therefore, further research is needed to improve the accuracy of AI-based measurements for these variables.

In this study, we included C2-7 SVA and C7 SVA based on distance information, which is different from previous studies that focused only on angles, to present findings on sagittal balance. The mean errors for C2-6 SVA and C7 SVA were 2.112 mm and 5.219 mm, respectively. These results are comparable to those obtained in another study that reported values of 1.1 mm and 1.9 mm for the same variables [19].

In addition to accuracy, the time required to obtain spinopelvic parameter measurements is also important. Our study found that the AI system was significantly faster than specialists in obtaining these measurements, with the AI system obtaining measurements in a matter of seconds compared to several minutes per patient for specialists.

The reduced time required for obtaining measurements by AI could have significant implications for clinical practice, particularly in terms of providing more timely diagnosis and treatment. AI-based systems have the potential to reduce the workload of specialists and increase efficiency in the diagnostic process. However, it is important to note that the use of AI-based systems should not replace specialists, but rather be viewed as a complementary tool to aid them in their decision-making process.

In comparison to previous studies, our AI detection and segmentation models achieved high accuracy in detecting and segmenting spinal structures in whole-spine lateral radiographs. For instance, Fujimori et al. reported an AP score of 0.6 for detecting vertebral corners in cervical spine radiographs using Faster R-CNN, while our best model achieved an AP score of 0.75 for detecting spinal structures in whole-spine lateral radiographs [17].

Comparing our results with previous studies, we found that our success rates were generally comparable or even better. For example, a previous study reported a success rate of 86.2% for angle measurements using deep learning-based segmentation on lumbar lordosis from X-rays, while our study achieved an overall success rate of 87.5%. However, it should be noted that the evaluation criteria and methodologies used in different studies may differ, making direct comparisons challenging.

In this study, we evaluated the success rate of angle measurement based on the segmentation results using both validation and test datasets. We applied an IoU threshold of 0.80 and used the hidden layers based on the weights of the best AP score of segmentation. Our results showed that three validation cases and seven test cases failed on all models.

In terms of the cases that failed on all models, further investigation is needed to identify the causes of the failures and improve the performance of the models. One possible reason could be the presence of anatomical variations or abnormalities that were not well represented in the training data. Future studies could consider expanding the training data set to include a wider range of anatomical variations and abnormalities to improve the models’ performance in these cases.

In this study, we developed a user-friendly program using the Tkinter library to analyze spinal structures in whole-spine lateral radiographs. Our trained model accurately predicted masks for the vertebrae, sacrum, and femur, and we used the OpenCV library to extract the two upper corners for spinopelvic parameter calculations. To assist users in identifying abnormalities in the measurements, we used a color-coded system that displayed results in blue for normal range and in red for outside the normal range [20,21]. Furthermore, our program displayed specific measurement results for each parameter and allowed users to save the results for future reference. Our color-coded system and specific measurement results display also add to the user-friendliness of our program. The use of the Tkinter library in our program also provides compatibility across multiple platforms, making it accessible to a wider range of users.

First, our study is limited by the exclusion of radiographs with implants and vertebral anomalies such as lumbosacral transitional vertebra, which was necessary for the development of a simple and intuitive model. However, including images with implants or more complex images in future research would be beneficial to improve the model’s accuracy and generalizability.

Second, the AI model was trained solely on X-rays from one institution, and this may limit its generalizability to other institutions with different settings and patient populations. Future research that includes data from multiple institutions could address this limitation.

Third, segmentation errors in the shoulder girdle area may result in inaccuracies in multiple parameters, particularly those involving the C7 or T1 body regions. Addressing this issue by refining the training model is necessary for future studies.

Lastly, our study only used lateral radiographs for spinal analysis, and while they are widely used in clinical practice, other imaging techniques like MRI may provide additional information that could improve the accuracy of the AI model. Incorporating multiple imaging modalities in future research may be necessary to validate the AI model’s accuracy.

The desirability of this study and the accompanying program hinges on the profound impact AI is set to have on spinal diagnostics and treatment. As mentioned in the limitation, radiographs with implants and vertebral anomalies were excluded from this study, so further evaluation is needed to obtain more accurate measurement results. Although AI-based evaluation still has many points to improve and has limitations, it will be a very efficient tool for time consuming tasks and tasks requiring accuracy.

## 5. Conclusions

The AI-based method for spinopelvic parameter extraction from whole spine long sagittal X-ray images shows promising results in terms of accuracy and speed. The method utilizes deep learning models to accurately detect and segment spinal structures, as well as a user-friendly interface to obtain spinopelvic measurements in a matter of seconds. The approach has the potential to revolutionize the field of spinal surgery by providing personalized treatment plans for each patient. However, the method’s limitations include the exclusion of radiographs with implants, training solely on X-rays from one institution, and potential segmentation errors in the shoulder girdle area. Future research that includes multiple imaging modalities and data from multiple institutions could improve the model’s accuracy and generalizability.

## Figures and Tables

**Figure 1 bioengineering-10-01229-f001:**
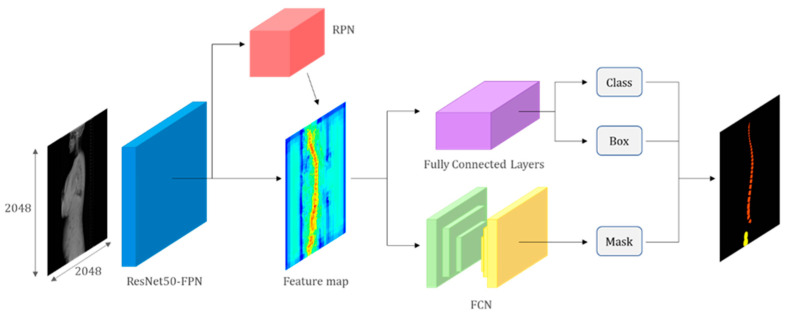
Artificial intelligence model architecture. The model comprises a backbone network (ResNet with FPN), an RPN, and a mask prediction network. The backbone network extracts feature from the input image, while the RPN predicts object regions. The mask prediction network generates binary masks for each region. The model is trained using R-CNN and RPN loss functions. The figure provides a concise overview of the model’s components and the training pipeline.

**Figure 2 bioengineering-10-01229-f002:**
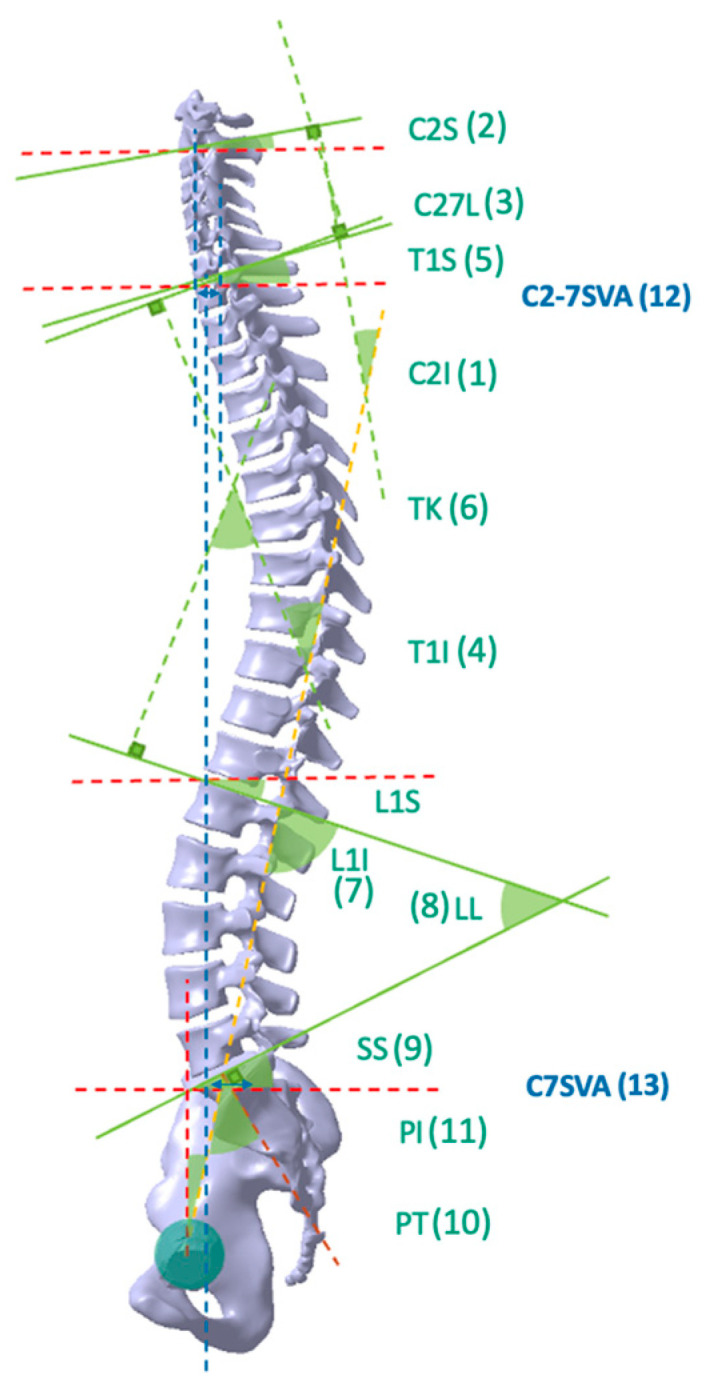
Illustration of key parameters in spinal sagittal alignment. The horizontal alignment is denoted by a red dashed line, while the vertical alignment is indicated by a blue dashed line. The brown dashed line represents a line that bisects the first sacral body upper endplate and is perpendicular to the endplate. The yellow dashed line signifies the line connecting the midpoint of the sacral body upper endplate and the center of the femur head. Lastly, the green lines depict the parallel lines corresponding to each endplate. 1. Cervical 2 Incidence (C2I); 2. Cervical 2 Slope (C2S); 3. Cervical 2 to 7 Lordosis (C27L); 4. Thoracic Incidence (T1I); 5. Thoracic 1 Slope (T1S); 6. Thoracic Kyphosis (TK); 7. Lumbar 1 Incidence (L1I); 8. Lumbar Lordosis (LL); 9. Sacral Slope (SS); 10. Pelvic Tilt (PT); 11. Pelvic Incidence (PI); 12. Cervical 2 to 7 Sagittal Vertical Axis (C2-7 SVA); 13. Cervical 7 Sagittal Vertical Axis (C7 SVA).

**Figure 3 bioengineering-10-01229-f003:**
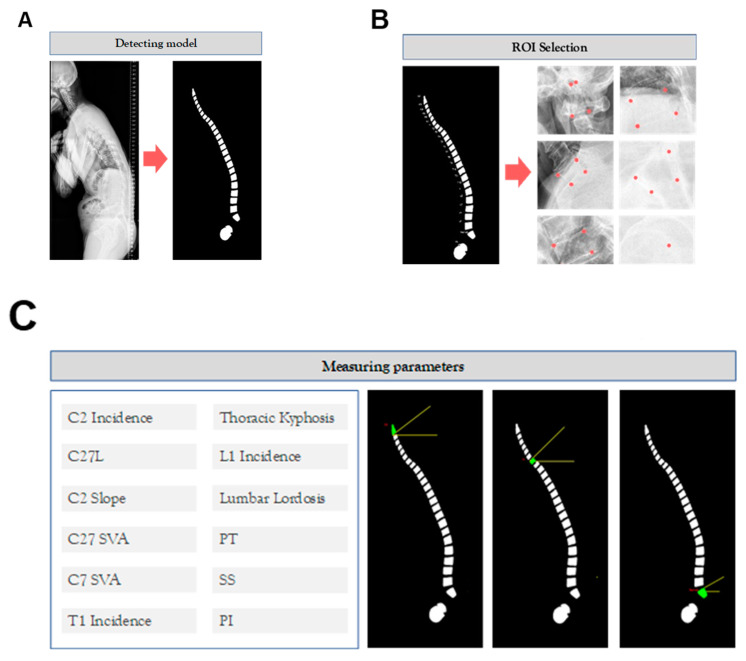
Graphical user interface (GUI). (**A**) The GUI incorporates a detection model that utilizes a pre-trained model to identify vertebrae, sacrum, and femur structures, generating corresponding masks. (**B**) Within the GUI, users can interactively select regions of interest (ROIs) within the detected structures, facilitating focused analysis. (**C**) To measure relevant parameters, the GUI approximates the shape of each generated mask and identifies the two upper corner points using the OpenCV library, enabling accurate parameter measurement.

**Figure 4 bioengineering-10-01229-f004:**
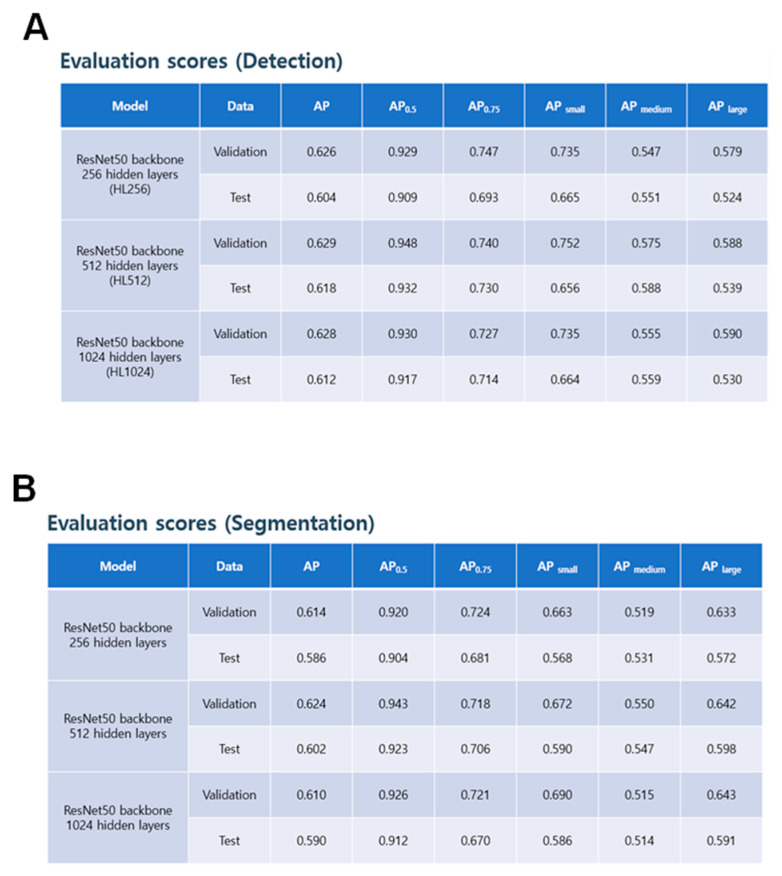
AI detection and segmentation. (**A**) The performance of the detection model was evaluated using AP scores for the validation and test datasets. (**B**) The segmentation model performance was assessed using AP scores for the validation and test datasets. (**C**) Visual representations of the input images, ground truth, and predicted masks for the detected structures.

**Figure 5 bioengineering-10-01229-f005:**
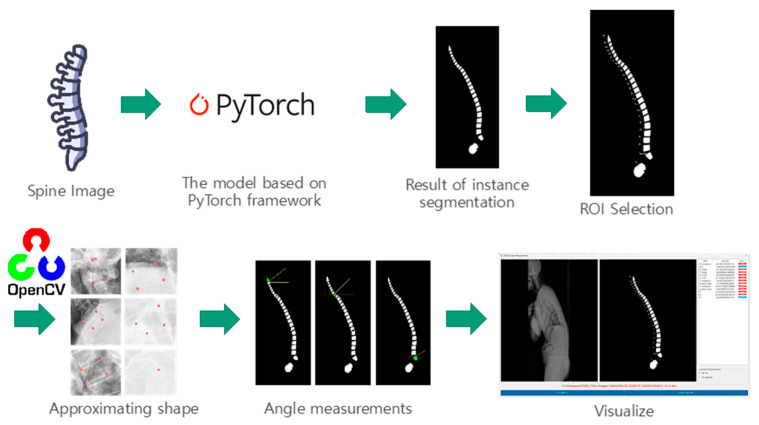
This figure showcases a program developed using the Tkinter library for spinopelvic parameter extraction. The program allows users to open an image, which is then analyzed using a trained model to predict masks for vertebrae, sacrum, and femur. The shape of the masks is approximated, and OpenCV is used to obtain the two upper corners for precise parameter measurement. Results are displayed with a color-coded system to indicate normal or abnormal ranges. Specific measurement values are provided, and users can save the results. The program achieves efficient parameter extraction, taking an average of 3.8 s.

**Table 1 bioengineering-10-01229-t001:** Comparison of measurements between Specialist vs. Resident and Specialist vs. Al.

Comparison of Measurements	C2I	C27L	C2S	T1S	C2-7 SVA	C7 SVA	T1I	TK	L1I	LL	PT	SS	PI	Time (s)
Specialist vs. Resident—Correlation	0.672	0.861	0.836	0.656	0.962	0.832	0.583	0.641	0.871	0.623	0.717	0.291	0.492	0.57
Specialist vs. Resident—R2	0.451	0.741	0.698	0.431	0.926	0.692	0.34	0.411	0.759	0.388	0.515	0.085	0.242	0.325
Specialist vs. Resident—Mean Error	7.98	6.204	5.143	6.49	2.282	9.594	8.816	8.49	6.143	10.327	5.918	8.286	9.429	184.531
Specialist vs. Resident—STD of Error	6.498	3.959	3.362	4.59	2.862	16.507	6.886	5.588	5.067	7.372	4.526	7.682	6.402	143.664
Specialist vs. AI—Correlation	0.92	0.955	0.913	0.836	0.978	0.922	0.848	0.794	0.96	0.79	0.844	0.685	0.794	0.258
Specialist vs. AI—R2	0.847	0.912	0.833	0.699	0.956	0.85	0.72	0.631	0.921	0.623	0.713	0.469	0.631	0.066
Specialist vs. AI—Mean Error	4.112	3.625	3.806	3.773	2.112	5.219	5.583	6.484	3.703	8.789	3.577	6.036	5.689	736.245
Specialist vs. AI—STD of Error	2.88	2.423	3.266	2.812	2.005	11.806	4.335	4.634	2.844	5.739	3.578	4.299	5.249	112.654

It presents the correlation, R-squared value, mean error, and standard deviation of the error for each of the 13 different measurements (C21, C27L, C2S, T15, C2-7 SVA, C7 SVA, T11, TK, L11, LL, PT, SS, and Pl) and the time taken in seconds. The table highlights the performance of Al compared to medical professionals, offering insights into the accuracy and potential benefits of using Al for these specific measurements.

**Table 2 bioengineering-10-01229-t002:** Comparing time for angle measurements.

	Mean (Second)	Mean Error	Standard Error
Resident	904.15	204.97	28.27
Specialist	732.4	105.1	16.1
AI	3.73	0.73	0.1

**Table 3 bioengineering-10-01229-t003:** Intraclass Correlation Coefficient (ICC).

Variable	Specialist vs. Resident	CI95%	*p*	Specialist vs. AI	CI95%	*p*
	ICC			ICC		
C2I	0.67	[0.48, 0.80]	<0.01	0.92	[0.86, 0.95]	<0.01
C2S	0.83	[0.70, 0.90]	<0.01	0.9	[0.83, 0.94]	<0.01
C27L	0.85	[0.75, 0.92]	<0.01	0.95	[0.92, 0.97]	<0.01
T1I	0.58	[0.36, 0.74]	<0.01	0.84	[0.73, 0.91]	<0.01
T1S	0.61	[0.36, 0.77]	<0.01	0.83	[0.72, 0.9]	<0.01
TK	0.62	[0.42, 0.77]	<0.01	0.74	[0.45, 0.87]	<0.01
L1I	0.87	[0.78, 0.92]	<0.01	0.95	[0.92, 0.97]	<0.01
LL	0.62	[0.42, 0.77]	<0.01	0.76	[0.57, 0.86]	<0.01
SS	0.27	[−0.0, 0.50]	0.027	0.62	[0.34, 0.79]	<0.01
PT	0.69	[0.49, 0.82]	<0.01	0.85	[0.74, 0.91]	<0.01
PI	0.49	[0.24, 0.68]	<0.01	0.78	[0.63, 0.87]	<0.01
C2-7 SVA	0.96	[0.92, 0.98]	<0.01	0.98	[0.96, 0.99]	<0.01
C7 SVA	0.79	[0.65, 0.88]	<0.01	0.92	[0.86, 0.95]	<0.01
Time(s)	0.31	[−0.07, 0.61]	<0.01	0	[−0.01, 0.01]	0.491

**Table 4 bioengineering-10-01229-t004:** Intraclass Correlation Coefficient (ICC): Bland–Altman plot.

Variable	Specialist vs. Resident	Specialist vs. AI
C2I	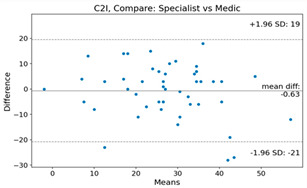	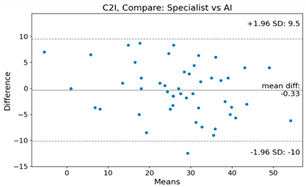
C2S	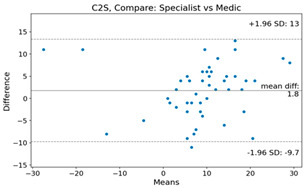	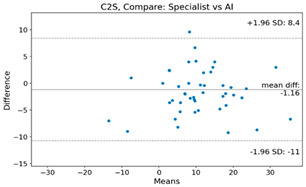
C27L	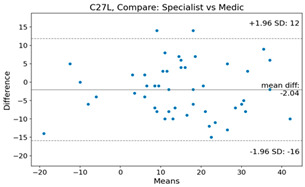	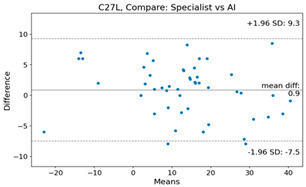
T1I	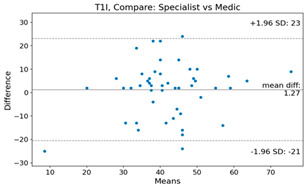	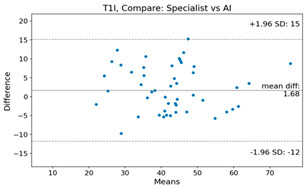
T1S	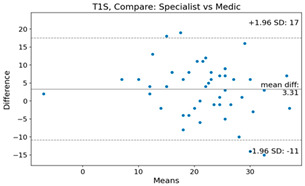	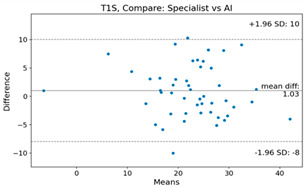
TK	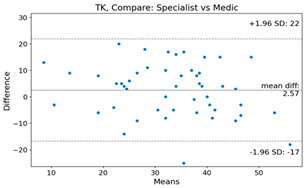	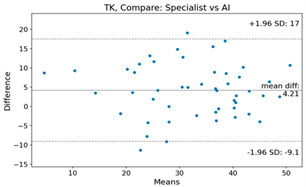
L1I	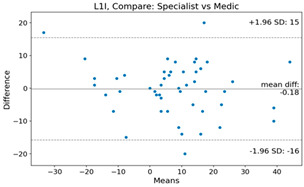	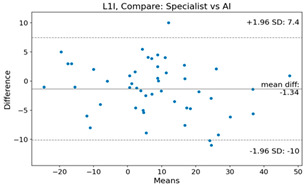
LL	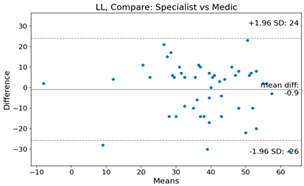	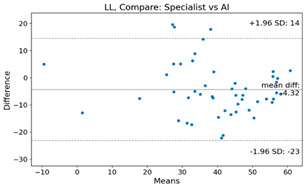
SS	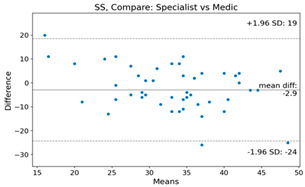	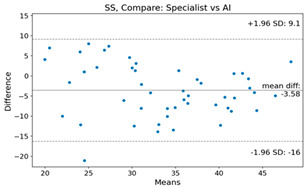
PT	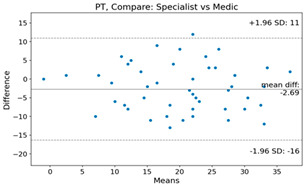	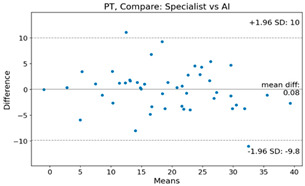
PI	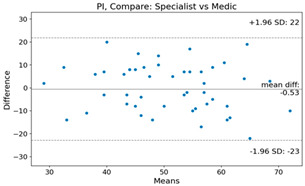	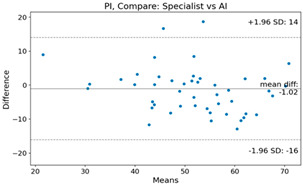
C2-7 SVA	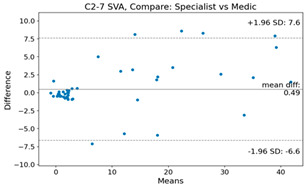	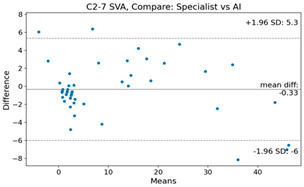
C7 SVA	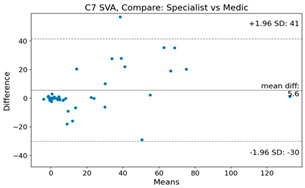	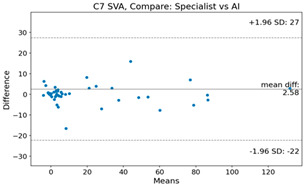
Time (s)	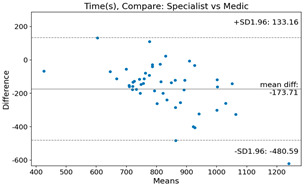	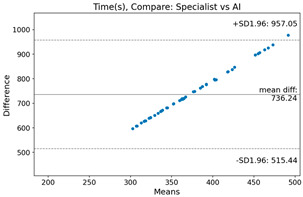

Represents the level of agreement between traditional medical practices and artificial intelligence (AI) measurements for various variables. A value of 1 indicates perfect agreement, while 0 indicates lack of agreement.

## Data Availability

Not applicable.

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
