# Peer review of "AI-Driven Segmentation and Automated Analysis of the Whole Sagittal Spine from X-ray Images for Spinopelvic Parameter Evaluation"

_bioengineering, 2023, doi:10.3390/bioengineering10101229_

Round 1
Reviewer 1 Report
Interesting article on a little covered topic.
The article is about AI-Driven Segmentation and Automated Analysis of the Whole Sagittal Spine from X-ray Images. The primary aim was to assess whether the proposed algorithm for automated whole-spine-pelvic detection and segmentation is good and accurate. There are few related articles. Well-written abstract. Introduction: well written, extensive literature review, well formulated research hypotheses. Please add clear study objectives. Materials and methods clearly and extensively presented. Clear, legible and nice tables and figures.
The results are well and widely presented. Clear, legible and nice tables and figures. The discussions should be improved by adding more arguments about the desirability of this program and study. Please add in the discussion that further research on this topic is needed and planned.
The references are appropriate.
Author Response
Response 1: Thank you for pointing this out. We agree with this comment. The content of the introduction has been added and supplemented to clarify the study. The revised contents can be found in page number 2, marked in red. “ If successful, the proposed AI algorithm could revolutionized the field of orthopedic sur-gery and spinal disorder diagnosis. It will make assessments more accurate, consistent, and efficient. This advancement can lead to better patient outcomes, fewer surgical com-plications, and a more streamlined process for medical professionals in evaluating and treating spinal disease.” The discussions are improved by adding arguments about the desirability of this study. The revised contents can be found in page number 16, marked in red. “The desirability of this study and the accompanying program hinges on the profound impact AI is set to have on spinal diagnostics and treatment. As mentioned in the limita-tion, radiographs with implants and vertebral anomalies were excluded from this study, so further evaluation is needed to obtain more accurate measurement results. Although AI-based evaluation still has many points to improve and has limitations, it will be a very efficient tool for time consuming tasks and tasks requiring accuracy.”
|

Reviewer 2 Report
This is a well builded and written paper dealing with an interesting field in medical progress such as artificial intelligence. Results are well reported and clearly show the utility of AI in clinical practice with great utility in time saving and measurement accuracy.
In the introduction "Although numerous studies... emphasizing the significance of malalignment factors, such as spinal and pelvic parameters [2]" this concept should be well expressed by recent review "doi: 10.1016/j.wnsx.2023.100162. I suggest this reference to be added to bibliography
In the discussion, I do not completely agree with the sentence "Deep mining algorithms can compare patients’ spinal alignment data to identify patterns associated with positive and negative prognoses, leading to improved patient outcomes. In this way, AI technology has the potential to revolutionize the field of spinal surgery and provide more effective and personalized treatment options." AI can contribute to pathology pattern and outcome comprehension through high volume data analysis but the revolution of spinal surgery and personalized surgery should be based on clinical expertise and technical evolution. This aspect, I think, should be remarked.
Author Response
Thank you very much for taking the time to review this manuscript. Please find the corresponding revisions highlighted changes in the re-submitted files.
Response 2: Agree. We have, accordingly, revised that paragraph in discussion to emphasize this point. The revised contents can be found in page number 14-15, marked in red.
Also, we updated the bibliography. “Although numerous studies have investigated treatment and prevention strategies for spinal disorders, recent research has shifted its focus towards the role of spinal alignment in these conditions, emphasizing the significance of malalignment factors, such as spinal and pelvic parameters [2,3].”
“While the role of AI in spinal diagnostics is undeniably transformative, it is essential to view it as a part the broader diagnostic tool, one that complements, not replaces, clinical expertise. As the field of spinal care evolves, the synergy between AI-derived insights and hands-on clinical experience will make better patient prognosis.”
Reference page number 17-18 3. Tartara, F.; Garbossa, D.; Armocida, D.; Di Perna, G.; Ajello, M.; Marengo, N.; Bozzaro, M.; Petrone, S.; Giorgi, P.D.; Schiro, G.R.; et al. Relationship between lumbar lordosis, pelvic parameters, PI-LL mismatch and outcome after short fusion surgery for lumbar degenerative disease. Literature review, rational and presentation of public study protocol: RELApSE study (registry for evaluation of lumbar artrodesis sagittal alignEment). World Neurosurg X. 2023, 18, 100162. https://doi.org/10.1016/j.wnsx.2023.100162.
|

Reviewer 3 Report
The authors present an interesting study in which an AI-based tool is developed in order to segment vertebral bodies and the proximal femur and calculate parameters related to spinal alignment. The performance of the automated tool was compared to that of orthopedic specialists, with the tool achieving results closer to the specialists than that of trainees. Overall, the manuscript has high scientific merit, with only a few areas of clarification required as noted below.
1. Recommend consistent use of the term “resident” or “medic” throughout the paper since the term “resident” is used in the abstract and Methods section 2.6, while “medic” is used thereafter. Furthermore, the experience level and instructions provided to the trainees should be mentioned.
2. Please mention whether patients with vertebral anomalies such as lumbosacral transitional vertebra were included in the study and how these cases were handled.
3. It is unusual to report p = 0 as in Table 3. The guidelines cited below recommend the following: “The precise p-value should be reported rather than stating that it is less than the level of significance or that it is insignificant. Generally, it is acceptable to report p-values to two decimal places (round to the nearest hundredth) when greater than 0.01; three decimal places (round to the nearest thousandth) when less than 0.01. If a p-value is quite small, then it is acceptable to report it as p-value < 0.001.”
Ou FS, Le-Rademacher JG, Ballman KV, Adjei AA, Mandrekar SJ. Guidelines for Statistical Reporting in Medical Journals. J Thorac Oncol. 2020 Nov;15(11):1722-1726. doi: 10.1016/j.jtho.2020.08.019. Epub 2020 Aug 25. PMID: 32858236; PMCID: PMC7642026.
Author Response
Thank you very much for taking the time to review this manuscript. Please find the corresponding revisions highlighted changes in the re-submitted files.
Response 3: Agree. We have, accordingly, revised as you mentioned. We change the terms from “medic” to “resident” to maintain the consistent use of the term. Page 9, line 314 “Table 1. Comparison of measurements between Specialist vs Resident and Specialist vs Al.” “Specialist vs Resident - Correlation Specialist vs Resident - R2 Specialist vs Resident - Mean Error Specialist vs Resident - STD of Error”
Page 10, line 320 “Based on the results presented in the table, the comparison between Specialist vs Resident and Specialist vs AI demonstrates the potential effectiveness of utilizing AI for specific measurements.” Page 10, line 324 “For Specialist vs Resident, the highest R-squared value was 0.926 for the C2-7 measurement, and the lowest was 0.085 for the PT measurement.” Page 10, line 331 “Taking the correlation coefficient as an example, the PT parameter shows a value of 0.844 for Specialist vs AI and a lower value of 0.717 for Specialist vs Resident.” Page 10, line 334 “Similarly, the R-squared value for the SS parameter is higher for Specialist vs AI at 0.631 compared to 0.325 for Specialist vs Resident.” Page 10, line 336 “In terms of MAE, the PI parameter has a value of 5.689 for Specialist vs AI, while it is Significantly higher at 9.429 for Specialist vs Resident.” Page 10, line 337 “This reveals that the AI has a lower average error compared to the Resident when compared to the Specialist.” Page 10, line 339 “Lastly, the STD of AE value for the SI parameter is lower for Specialist vs AI at 4.299, as opposed to 7.682 for Specialist vs Resident”
The experience level and instructions provided to the trainees should be mentioned. Page 6, line 222 “To evaluate the agreement between measurers (specialists, residents-3rd year trainee, and AI), the Intraclass Correlation Coefficient (ICC) was used as an indicator.”
We excluded the vertebral anomalies in this study. This is the one of the limitation, so we add that contents in discussion. The revised contents can be found in page number 16, marked in red. “First, our study is limited by the exclusion of radiographs with implants and vertebral anomalies such as lumbosacral transitional vertebra, which was necessary for the development of a simple and intuitive model.”
We revised the p-value in table 3. As you mentioned, p = 0 is unusual, therefore we revised that expression “p < 0.01” to show acceptable results. |
Table 3. Intraclass Correlation Coefficient (ICC)
variable |
Specialist vs Resident |
  |
  |
Specialist vs AI |
  |
  |
  |
ICC |
CI95% |
p |
ICC |
CI95% |
p |
C2I |
0.67 |
[0.48 0.80 ] |
< 0.01 |
0.92 |
[0.86 0.95] |
< 0.01 |
C2S |
0.83 |
[0.70 0.90] |
< 0.01 |
0.9 |
[0.83 0.94] |
< 0.01 |
C27L |
0.85 |
[0.75 0.92] |
< 0.01 |
0.95 |
[0.92 0.97] |
< 0.01 |
T1I |
0.58 |
[0.36 0.74] |
< 0.01 |
0.84 |
[0.73 0.91] |
< 0.01 |
T1S |
0.61 |
[0.36 0.77] |
< 0.01 |
0.83 |
[0.72 0.9 ] |
< 0.01 |
TK |
0.62 |
[0.42 0.77] |
< 0.01 |
0.74 |
[0.45 0.87] |
< 0.01 |
L1I |
0.87 |
[0.78 0.92] |
< 0.01 |
0.95 |
[0.92 0.97] |
< 0.01 |
LL |
0.62 |
[0.42 0.77] |
< 0.01 |
0.76 |
[0.57 0.86] |
< 0.01 |
SS |
0.27 |
[-0.0 0.50] |
0.027 |
0.62 |
[0.34 0.79] |
< 0.01 |
PT |
0.69 |
[0.49 0.82] |
< 0.01 |
0.85 |
[0.74 0.91] |
< 0.01 |
PI |
0.49 |
[0.24 0.68] |
< 0.01 |
0.78 |
[0.63 0.87] |
< 0.01 |
C2-7 SVA |
0.96 |
[0.92 0.98] |
< 0.01 |
0.98 |
[0.96 0.99] |
< 0.01 |
C7 SVA |
0.79 |
[0.65 0.88] |
< 0.01 |
0.92 |
[0.86 0.95] |
< 0.01 |
Time(s) |
0.31 |
[-0.07 0.61] |
< 0.01 |
0 |
[-0.01 0.01] |
0.491 |
